# Laparogastroscopy—A Transgastric Laparoscopic Approach for Malignant Esophageal Stenosis

**DOI:** 10.3390/healthcare11060815

**Published:** 2023-03-09

**Authors:** Alexandra Delia Lupu-Petria, Alexandru Dan Sabau, Dragos Serban, Tiberiu Trotea, Ionela Maniu, Dan Sabau

**Affiliations:** 1Faculty of Medicine, “Lucian Blaga” University Sibiu, 550169 Sibiu, Romania; 2Department of Surgery, Emergency County Hospital, 550245 Sibiu, Romania; 3Faculty of Medicine, “Carol Davila” University of Medicine and Pharmacy, 020021 Bucharest, Romania; 4Fourth Department of Surgery, University Emergency Hospital, 050098 Bucharest, Romania; 5Mathematics and Informatics Department, Faculty of Sciences, Research Center in Informatics and Information Technology, “Lucian Blaga” University Sibiu, 550025 Sibiu, Romania; 6Research Team, Clinical Pediatric Hospital, 550166 Sibiu, Romania

**Keywords:** laparogastroscopy, esophageal, cancer, surgery, palliation

## Abstract

This paper presents the laparogastroscopy procedure, a mini-invasive, palliative method as an alternative to gastrostomy to be recommended by gastroenterologists. Laparogastroscopic stenting with endoluminal transtumoral drilling solves the problem of oral nutrition in patients with unresectable esophageal cancer, avoiding percutaneous feeding. The results of this technique are presented in a retrospective analysis of a study group of 63 patients with advanced esophageal carcinoma admitted between January 2015 and December 2020 at Department of General Surgery of Emergency County Hospital Sibiu, Romania, in terms of post-operative morbidity and mortality. The type of stents used were Pezzer prostheses (48.6%), silicone prostheses (31.9%), and self-expanding metal stents (6.9%). Eight patients (12.7%) had fistulas (at admission to the clinic), which were successfully sealed. Post-operative dysphagia was absent in most patients and minimal in 16.6% of patients, so all patients could initiate oral feeding, improving their nutritional status. The average length of hospitalization for all patients was 9.22 ± 5.05 days. The most frequent local complications were restenosis (9.5%), stent displacement (7.9%), and bleeding (4.8%). The mean survival time was 10.75 ± 15.72 months. Laparogastroscopic stenting could be a valuable alternative in palliative esophageal cancer surgery, improving the quality of life and nutritional status in patients unsuitable for endoscopic stenting.

## 1. Introduction

Esophageal cancer is the seventh most common cancer in the world, accounting for 3% of all cancers [1,2]. It is one of the most severe forms of digestive cancer, with a very high mortality rate. In industrialized countries, esophageal neoplasm is the sixth leading cause of cancer death.

Esophageal neoplasm is unfortunately diagnosed, in most cases, in an advanced stage, because patients do not usually show any symptoms until the advanced stages of the disease, when dysphagia occurs. Tumors are often unresectable due to locoregional invasion, increased size, metastasis, patient refusal of surgical treatment, and refusal of radio–chemotherapy [3,4], or due to previous surgery that prevents esophagectomy in an earlier stage. Locoregional invasion can often be complicated by the presence of fistulas to the trachea or fistulas to the bronchus. For these reasons, the main therapeutic option remains palliation [2,5,6,7].

Self-expanding metal stents (SEMSs) were introduced in the early 1980s and are now considered the gold standard in the palliation of advanced esophageal cancer (std III and IV) [8].

The use of endoscopically self-expanding metal prosthesis is a great advantage, as it is a modern method that provides the patient with the possibility of oral feeding and integration into society, thus increasing the quality of life and increasing overall survival from 1.2 months, on average, without endoscopic intervention to 9–12 months [9]. Unfortunately, endoscopy does not treat pharyngoesophageal and esophageal placements or narrow stenosis that do not make the passage of the orthograde guidewire possible [9,10].

The aim of this paper is to describe a mini-invasive, palliative method, i.e., laparogastroscopic stenting with endoluminal transtumoral drilling. This method represents a technical, biological, and social alternative to gastrostomy, jejunostomy, or pharyngostomy, which are procedures that are also used in open or laparoscopic surgery in oncological patients. Laparogastroscopy is a solution for stenting unresectable esophageal cancer in (1) cases that cannot be endoscopically resolved due to the technical impossibility of crossing the stenosis (the stenosis is complete and does not make the orthograde passage of the guidewire possible), (2) tumor stenosis located less than 2 cm from the upper esophageal sphincter, and (3) avoidance of the esogastric junction.

## 2. Materials and Methods

### 2.1. Laparogastroscopy Technique Description

Stenting with the laparogastroscopic approach involves laparoscopy, abdominal lesion inventory/evaluation, left subcostal detection, and exteriorization of the anterior gastric wall (near the great curvature) with fixation to the tegument in four cardinal points. Next, we endogastrically insert the trocar to visualize the stomach, cardia, and inferior esophagus (Figure 1a). The esophageal visualization is extended, when necessary, using the telescope-in-telescope maneuver (Figure 1b). The stenotic area is retrograde catheterized, with externalization of the catheter at the oral level. The catheter is anchored to a progressive stent system (in diameter) for tumor drilling or scar dilatation with the collection of scraped biopsy material and oral traction of the prosthesis to the stenotic area, where due to the funnel device, it tends to self-lock, thus achieving hemostasis or coverage of the fistula to the trachea or fistula to the bronchus. The devices used for the surgical intervention are presented in Figure 2a–g. 

#### The Type of Stents

Different type of stents were used in the study. Uncovered self-expandable metal stents (Ultraflex stent (Boston Scientific, MA, USA); length of 120/150 mm and diameter of 23/18 mm; Figure 3a), silicone stents (Demed, Mikolow, Poland or CERTEX, Romania; Figure 3b), and Pezzer catheters (Figure 3c) were used.

In high tumor placements, the prosthesis can be fixed to the posterior wall of the hypopharynx. In esogastric placements, the stent can be endogastrically or transparietogastrically fixed to the small curvature of the stomach (Figure 4a–e). In the endogastric case, a curved needle is used to fix the prosthesis to the gastric wall, without penetrating the serosa (Figure 5a). The wire is knotted inside the stomach using a single-port instrument. In the transparietogastric case, a straight needle is used, penetrating the gastric wall from the inside to the outside (Figure 5b). The needle is endoperitoneally visualized and reintroduced into the stomach with a separate penetration, with the endogastric knotting of the wire.

### 2.2. Rendezvous Technique Description

The need for the good visualization of the proximal and distal esophageal lumina, the proximal and distal tumor boundaries, and the stenotic orifice led us to use a mixed technique combining intra-operative endoscopy and laparogastroscopy, which are both minimally invasive methods. The technique involves a partnership between gastroenterology and surgery and can be used at the extreme level of the digestive tract. The indirect rendezvous technique involves a laparogastroscopic encounter and the effects of endoscopy (guidewire), and the direct rendezvous technique involves the simultaneous presence of an endoscope and a laparogastroscope at the tumor site. In this procedure, the catheterization of the stenotic orifice is visible, and the stent can be precisely positioned in relation to the stenosis (which is sometimes shorter and sometimes longer); the need for two telescopic stents can be accurately assessed.

As the working instrument, we used a telescope with a working channel of 6 mm, a single port of 20 mm in diameter with three holes, a telescope of 450 mm in length and 5 mm in diameter inserted in a telescope of 10 mm in diameter and 280 mm in length, and laparoscopic instruments for the approach and fixation of the prosthesis (Figure 2a–c,e).

The stent type was selected on an individual case basis. For hard cartilaginous tumors, we used rigid plastic stents (Pezzer; silicone), thus avoiding the transformation of self-expandable metal stents into clepsydra (shaped) stents. Rigid plastic stents have good strength, almost immediately relieve dysphagia, and prevent tumor ingrowth. In the case of long tumors in the axis of the esophagus, Pezzer stents are preferred because their length can be intra-operatively adapted. We also used SEMSs, which are easier to fix than plastic or semirigid stents but are less stable. Uncovered and partially covered SEMSs are at high risk of restenosis, and the covered ones are prone to migration. Another criterion for not using SEMSs is their availability and higher financial costs in comparison with plastic or semirigid stents.

### 2.3. Patient Selection in the Study Group

This retrospective analysis of hospital records included data of a consecutive group of patients with advanced esophageal carcinoma treated between January 2015 and December 2020 at Department of General Surgery of Emergency County Hospital Sibiu, Romania. All these patients presented contraindications to oncological radical surgical procedures (esophagectomy) and firm indication of gastrostomy, jejunostomy, and pharyngostomy. Demographic and clinical data, including age, sex, weight, dysphagia, dyspnea, chemotherapy/chemoradiation, stent migration, and complications, were evaluated.

The inclusion criteria for the study were upper, middle, and lower esophageal tumors; impossibility of endoscopic stenting; contraindication to esophagectomy; severe partial or total dysphagia; patients who had fistulas to the trachea or fistulas to the bronchus; patients with performance status (assessed according to Karnofsky score) more than or equal to 40 [11]; and pathological diagnosis of esophageal squamous cell carcinomas and adenocarcinoma. Patients on neoadjuvant chemotherapy were not excluded from the study.

The exclusion criteria were patients with altered general condition not allowing general anesthesia to be performed; patients who could be endoscopically stented; and patients with incipient esophageal cancers suitable for radical surgery (especially of the esogastric junction; Sievert I, II, and III).

### 2.4. Pre- and Post-Treatment Assessments

Unresectability was determined on the basis of chest radiography, abdominal ultrasound, computed tomography (CT) of the chest and upper abdomen, positron-emission tomography (PET), and endoscopy with endoscopic ultrasound (EUS) and endobronchial ultrasound (EBUS). Disease staging was based on the UICC classification [12,13]. Dysphagia is perceived by the patient as a difficulty in swallowing solid and/or liquid food. Clinically, dysphagia occurs when the esophageal lumen is more than 75% blocked (differentiated for polar strictures). Dysphagia was assessed according to the four-grade scale Ogilvie dysphagia classification [13], where grade 0—to be able to eat an ordinary diet; grade 1—to be able to eat solid food; grade 2—to be able to eat semi-solids; grade 3—to be able to eat liquid foods; and grade 4—to be able to eat nothing.

Patients diagnosed with a fistula in the course of esophageal or bronchogenic cancer were classified into four groups according to fistula location [14], where Type 1—fistula to the mediastinum; Type 2—fistula to the trachea; Type 3—fistula to the bronchus; and Type 4—fistula after stenting.

Dyspnea severity was assessed with a four-grade scale [14], where 0—less than 30% tracheal or/and bronchial stenosis, no dyspnea; 1—30–50% stenosis, dyspnea upon exercise; 2—50–70% stenosis, dyspnea during daily activities; and 3—more than 70% stenosis, dyspnea while resting.

The resumption of liquid feeding was performed post-operatively, when the post-anesthesia status made it possible. On the first post-operative day, a hydrolactosaccharide diet was adopted; then, starting from the third post-operative day, semi-solid feeding was stopped, specifying the necessity of its crushing. When switching to a semi-solid diet, the patient ingested a spoonful of olive oil before the meals to act as a lubricant and consumed a glass of fizzy drink (Coke) at the end of the meals for the entire life of the prosthesis. Vitamins A, C, and E, and Resveratrol with an antioxidant role also improved the quality of life of these patients, and so inevitably did oncoadjuvant therapy.

Post-operative complications were determined clinically (general condition, and assessment of dysphagia and dyspnea scores) and on the basis of chest radiography, computed tomography of the chest and abdomen if necessary, abdominal ultrasound, endoscopic ultrasound, and endoscopy in case of stent migration or restenosis of stent due to impacted food debris or tumor overgrowth that stenosed the prosthesis.

### 2.5. Statistical Analysis

Continuous characteristics were presented as means, standard deviations (SDs), and ranges, while categorical data were presented as percentages. Comparisons between independent groups (two or more) were performed using nonparametric (Mann–Whitney U and Kruskal–Wallis H) or parametric (Student’s *t*) tests in case of continuous variables, while for categorical data, chi-squared or Fischer’s exact test were used. The normality of the data was assessed using the Shapiro–Wilk test [15,16]. A *p*-value of 0.05 was considered statistically significant. Statistical analyses were performed with SPSS version 20 software (SPSS Inc., Chicago, IL, USA).

## 3. Results

Between 2015 and 2020, 72 patients with esophageal neoplasm in stage III/IV presented at the Surgery II clinic of Sibiu Emergency County Hospital. Of these seventy-two patients, nine patients were excluded from this analysis due to the following causes: one patient was out of surgical resources; two patients refused prosthesis; two patients had gastrostomy; one patient had gastrointestinal bleeding; and three patients required further investigation and treatment. The remaining 63 patients underwent esophageal stenting for malignant esophageal obstruction with laparogastroscopy and the rendezvous technique (13 of them were treated with the indirect rendezvous technique, and 3 of them were treated with the direct rendezvous technique, while the rest were treated with laparogastroscopy without the rendezvous technique) and were included in further analysis.

The age of the patients ranged from 36 to 95 years, with an average of 64.5 years, with most cases of esophageal cancer occurring in patients aged 56–75 years and the gender distribution being 87.5% male (the mean age of male patients was 63.55 years, and that of female patients, 61.6 years). The stenotic tumor was located in the lower esophagus in 42.9% of cases, in the middle esophagus in 36.5% of cases, and in the upper esophagus in 20.6% of cases. Pezzer prostheses were used in 48.6% cases; silicone prostheses, in 31.9% of cases; and SEMSs, in 6.9% of cases.

Eight patients (12.7%) had a fistula (at admission to the clinic), with seven being fistulas to the trachea and one being a fistula to the bronchus. Extreme malnutrition with total dysphagia was present in 60.3% of patients, with the remaining 39.7% of patients presenting selective dysphagia for solids and semi-solids and partially for liquids. Post-stenting dysphagia was absent in most patients, and only in 12 cases (in the immediate peri-operative period), the persistence of dysphagia for solids was encountered. It was transient a few days after changing eating habits and educating the patient to better grind food.

Generalized metastases occurred in 6 (9.5%) patients; pulmonary metastases, in 14 (22.2%) patients; liver metastases, in 5 (7.9%) patients; bone metastases, in 3 (4.8%) patients; intrathoracic adenopathy, in 35 (55.6%) patients; intra-abdominal adenopathy, in 3 (4.8%) patients; and laterocervical adenopathy, in 2 (3.2%) patients. The baseline characteristics of the study group are presented in Table 1.

A summary of post-stent complications is presented in Table 2. Retrosternal pain (more than 1 month) after stent placement was encountered in 60.3% of cases. There were no cases with massive hemorrhage during stent placement, but this complication developed post-placement in three (4.8%) patients (for one of them, this was fatal).

Migratory complications (migration into the stomach) of the prosthesis were present in five male patients, who required re-operation for stent removal and exchange or repositioning. Generally, these complications occur post-radio- and chemotherapy.

In total, six patients had restenosis; three of them had impacted food debris in the tumor that was resolved with endoscopic exploration and lavage; and three of them had tumor growth that stenosed the prosthesis.

We recorded 12 deaths among patients with prostheses, with the most common causes being cardiac causes, sepsis, pulmonary causes (pulmonary decompensation due to advanced esophageal neoplasm, aspiration pneumonia, or pulmonary abscess), restenting procedure, or complications due to comorbidities. The mean survival time of these patients was 10.75 ± 15.72 (0–50) months (Table 3).

The average length of hospitalization for all patients was 9.22 ± 5.05 days, with non-significant differences in the location of cancer (upper esophagus, 11.76 ± 6.62; middle esophagus, 8.13 ± 3.82; lower esophagus, 9.03 ± 4.89; *p* = 0.110) or prosthesis type (Pezzer prosthesis, 9.57 ± 5.12; silicone prosthesis, 9.09 ± 5.37; SEMS, 7.91 ± 2.89; *p* = 0.472).

## 4. Discussion

This paper presents the laparogastroscopy procedure, a mini-invasive, palliative method as an alternative to gastrostomy to be recommended by the gastroenterologist (when endoscopic stenting is impossible). Laparogastroscopic stenting with endoluminal transtumoral drilling solves the problem of oral nutrition in patients with unresectable esophageal cancer, thus avoiding percutaneous feeding. Laparogastroscopic stenting does not displace endoscopic stenting, which is the “gold standard”, but solves gastroenterological impossibilities and technical limitations (difficulties related to the endoscopic approach, visualization, and placement, and the impossibility to easily approach the esophageal poles, which endoscopists are reluctant to do because of possible complications that may arise later). Laparogastroscopy can be iteratively performed, resolving the removal of migrated prostheses and the repositioning of a new stent in the same session. The direct visualization of the peritoneal cavity with laparoscopic exploration is another advantage, as it allows more accurate staging to be performed by detecting visceral or peritoneal metastases that are often overlooked during pre-operative investigations.

Plastic prostheses are easier to place, tend to self-lock at the funnel, physically drill through the tumor (compared with flexible prosthesis, which can compress due to the advanced tumor process), help in hemostasis (in case of hemorrhage triggered by transthoracic drilling), and reduce tumor invasion. The migration of prostheses fitted at the gastric–mesial junction can be prevented by fixing plastic and SEMSs. The procedure was applied to a category of patients condemned by the advanced nature of the disease (stage IV esophageal cancer) and candidates for disabling methods (gastrostomy, jejunostomy, and feeding pharyngostomy). As a result of the procedure, patients preserved the ability to taste food, chew, eat in public or with their family, and avoid overly sophisticated methods of food preparation. Compared with PEG, our procedure makes oral nutrition possible for patients with stage IV cancer; thus, after stenting, laparogastroscopy improvements in metabolic status and comfort of life were observed in most patients. The mean dysphagia score improved from 3–4 before stenting to 0–1 after stenting.

Post-stent placement complications in this study were pain, restenosis, stent migration, and hemorrhage. Endogastric distal migration was resolved with reintervention with the laparogastroscopic removal of the prosthesis and the fitting of a new prosthesis. Typically, the patient did not notice the migration of the prosthesis, due to the very slow recovery of the stenosis, and so they were delayed in reporting this to their doctor. Prosthetic migration generally occurs in oncological cases after radiotherapy, or when the stenosis is destroyed by ischemia or is softened due to compression (prolonged decubitus). No patient encountered massive hemorrhage during stent placement. Post-operatively, after 3 weeks, one case with a tracheoesophageal fistula presented with cataclysmic bleeding in the trachea, resulting in death.

Other reports included the following complications with metallic stents: pain, hemorrhage, stent obstruction by tumor/food, stent migration, tumor ingrowth/overgrowth, restenosis, perforation, aspiration pneumonia, tracheoesophageal fistulas, and reflux esophagitis [7,17,18,19,20,21,22,23]. In study [17], which analyzed self-expanding metal stents and intraluminal radioactive stents for inoperable esophageal squamous cell carcinoma, the main complications following stent placement included pain (17.6% vs. 30.0%), massive hemorrhage (6.6% vs. 2.5%), stent migration (5.5% vs. 5%), and restenosis (4.4% vs. 5%). In a review article covering various aspects of self-expanding metallic stent placement for palliating esophageal cancer, SEMS complications included chest pain (up to 14% of cases), migration (between 7% and 75% of cases), a small amount of bleeding (in 5% of cases)/major bleeding (<1% of cases) [20,21], and perforation during or soon after SEMS placement (<1% of cases) [18,19,20]. Other studies reported pain with a rate between 18.7% and 74% [7,17,21,22]. In a study [23] aiming to compare the clinical outcomes following the placement of fully covered, self-expanding metallic stents (FCSEMSs) vs. partially covered, self-expanding metallic stents (PCSEMSs) for the palliative treatment of inoperable esophageal cancer, the reported complications were stent migration (up to 19.04% with FCSEMSs and up to 29.78% with PCSEMSs); stent obstruction by tumor/food (up to 23.81%/9.52% with FCSEMSs and up to 30.95%/13.33% with PCSEMSs); chest pain (up to approximately 19% in both cases) and bleeding (up to 8.33% with FCSEMS and up to 11.90% with PCSEMS). Our results regarding complications were similar to those of the endoscopic stenting procedure. Compared with stenting procedures, we had no perforation nor fistulas after stenting, because the use of laparogastroscopy in the rendezvous technique provides good visibility and added safety of catheterization to our method.

In case of bypass surgery, complications that frequently occurred in patients were suture insufficiency (42.7%), anastomotic leakage (8–40%), leakage in the remnant esophageal stump (15%), recurrent laryngeal nerve paralysis (11.8%), pneumonia (11.8%), abdominal abscess (11.8%), torsion of the gastric tube (5.9%), and surgical site infection (32%) [24,25,26,27]. Compared with open surgery, our method is mini-invasive, so post-operative complications, mortality, and hospitalization days are lower.

Stenting with laparogastroscopy has shown superior efficacy for the palliation of malignant dysphagia compared with photodynamic therapy, laser therapy, and esophageal bypass [28], improving patient nutritional status. In our study, dysphagia was absent in most patients and minimal in 16,6% of patients, so all patients could initiate oral feeding.

In our study, the average length of hospitalization was 9 days. The length of hospitalization depends on the fact that patients with stage IV esophageal neoplasm have an impaired general condition, present multiple comorbidities, and require additional surveillance [29,30].

Several studies reported that the mean survival in the surgical gastrostomy group was 3–4 months; in the bypass group, it was 3–9.7 months; in the endoscopic gastrostomy group, it was 4–7 months; in palliative resection, it was 7.8–13 months; in (NdYaG) laser treatment, it was 7.2 months; and in the stent group, it was 2.9–6 months [24,25,27,31,32,33,34,35,36,37,38]. Regarding survival time, our results with laparogastroscopic procedure are encouraging and comparable to the above-mentioned methods. Moreover, in our study group, there was one patient with survival time of 50 months, and long-term survival (of more than 3 years) has been mentioned in esophageal bypass surgery but not in endoscopic stenting [25,39,40,41,42,43].

There were also cases in which the laparogastroscopic method had its limits. Two patients with esogastric tumors required the digital drilling of the gastroesophageal lumen in open surgery due to the impossibility of the catheterization of the invisible tumor lumen under endoscopic, laparogastroscopic, or mixed approach conditions. One pharyngeal tumor location required pharyngostomy, and two others involved feeding jejunostomy. Two cases with post-operative recurrence after hypopharyngeal surgery had pharyngocutaneous fistulas. The prostheses located in the upper esophagus in three cases and in the thoracic esophagus in two cases were removed by coughing or vomiting.

## 5. Conclusions

From our point of view, laparogastroscopy (minimally invasive method) is recommended in cases of tumor overgrowth, with tumors that are inaccessible with endoscopy and for which open surgery has high costs. Additionally, (1) it decreases the length of hospitalization of patients, and consequently, hospitalization costs decrease [26]. (2) It is a technique with a medium learning curve, requiring competence in thoracic and abdominal surgery. (3) It avoids disabling surgical procedures used in open surgery or endoscopy (PEG (percutaneous endoscopic gastrostomy)) [38,44,45,46,47]. (4) Laparogastroscopy allows the assessment of the cancer stage to be performed with endoperitoneal visualization (during transtumoral drilling, biopsy material can be taken). (5) There is an increase in the quality of life of patients, as this technique allows them to taste food and, at the same time, gives them the opportunity not to socially isolate themselves. The rendezvous technique provides added safety for catheterization and the topographic placement of the prosthesis. In terms of stents, the hard compressible and impermeable plastic prostheses used in this method exclude the risk of tumor invasion of the lumen; funnel prostheses enable hemostasis to be achieved with compression at the same time as fitting; impermeable plastic prostheses enable tracheoesophageal or bronchoesophageal fistulas to be sealed; and plastic or semirigid stents have reduced costs. The limitations of the method are generated by the inability to catheterize the esophageal lumen.

Considerations related to the reservations or inefficiency of endoscopic therapy, post-operative evolution, the disabling nature of some of the interventions (gastrostomy and jejunostomy), or the excessive nature of other techniques (in relation to the size and evolutionary perspective), to which we add cosmetic reasons, led us to approach the esophagus with an original, innovative, minimally invasive surgical variant in oncological palliation. The method can also be used in post-caustic scarring or cardiospasm (achalasia) (to illustrate the versatility of the method, we present three cases in Appendix A).

Using the proposed technique, the undernourished, emaciated patient with esophageal stenosis (most frequently malignant), avoided or unresolved using endoscopic techniques, has a chance to prolong (months to years) the ability of oral feeding, with all its individual and social advantages. Our method is comparable with endoscopic stenting regarding complications, survival, and post-operative mortality (we had no perforations, no leakage, nor post-operative fistulas, and pre-operative fistulas were successfully sealed). It is a safe, effective, and efficient procedure and is an alternative to PEG, bypass surgery, or surgical gastrostomy.

## Figures and Tables

**Figure 1 healthcare-11-00815-f001:**
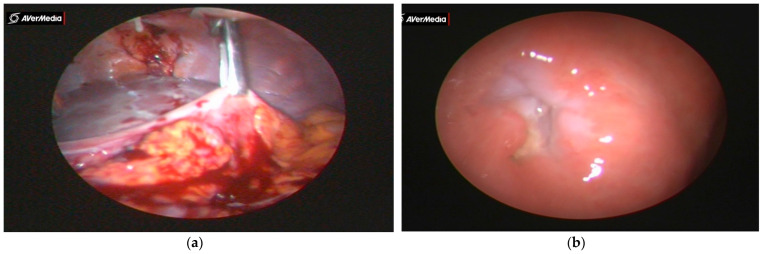
(**a**) Gastric anchorage with crocodile clamp and left subcostal eternalization and fixation. (**b**) Visualization of the cardia through which the telescope with the working channel and the elastic catheter are inserted in view of the cranial passage of the tumor and its oral extraction.

**Figure 2 healthcare-11-00815-f002:**
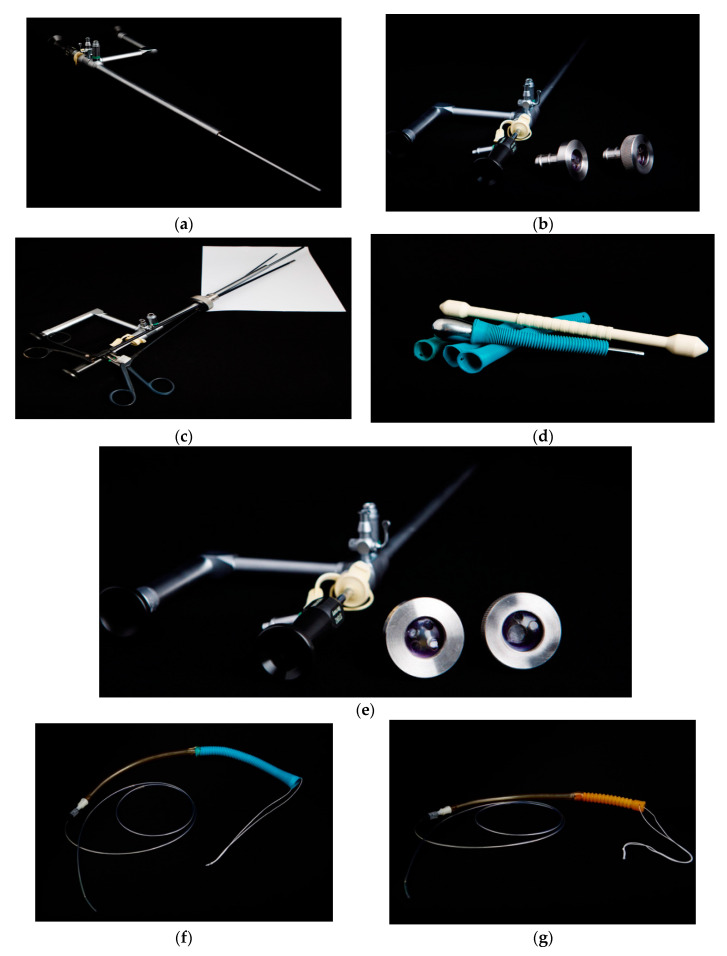
(**a**) Telescope with working channel. (**b**) Telescope with working channel with probe for retrograde catheterization (single-port system). (**c**) Auxiliary surgical instruments with single-port system. (**d**) Instruments used for the thermoregulation of stents. (**e**) Single-port system with 2 and 3 holes. (**f**) Anchoring assembly: tent-anchored oral fixation wire, stent, and intermediate segment crossed by endoluminal wire anchored by stent (upper pole) and lower-pole catheter. (**g**) Traction: complex proximal tension wire system, stent, intermediate tube and distal traction wire, and traction catheter.

**Figure 3 healthcare-11-00815-f003:**
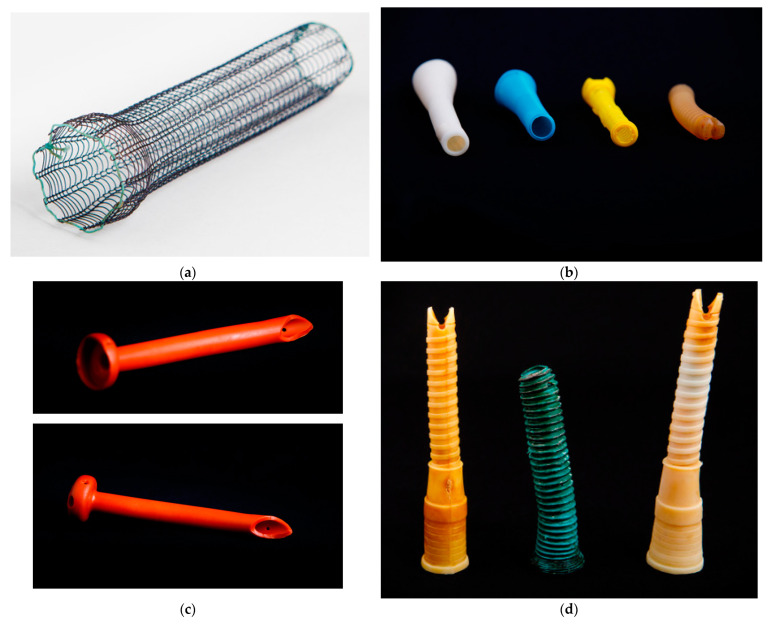
(**a**) Self-expanding metal stent. (**b**) Prosthesis with successive diameters. Stents used for esophageal stenting with laparogastroscopy—the original method used since 1996. (**c**) Pezzer prepared at funnel level and dimensionally adapted (3–18 mm in diameter) according to possibilities, 10–15 cm in length. (**d**) Migrated and extracted stents.

**Figure 4 healthcare-11-00815-f004:**
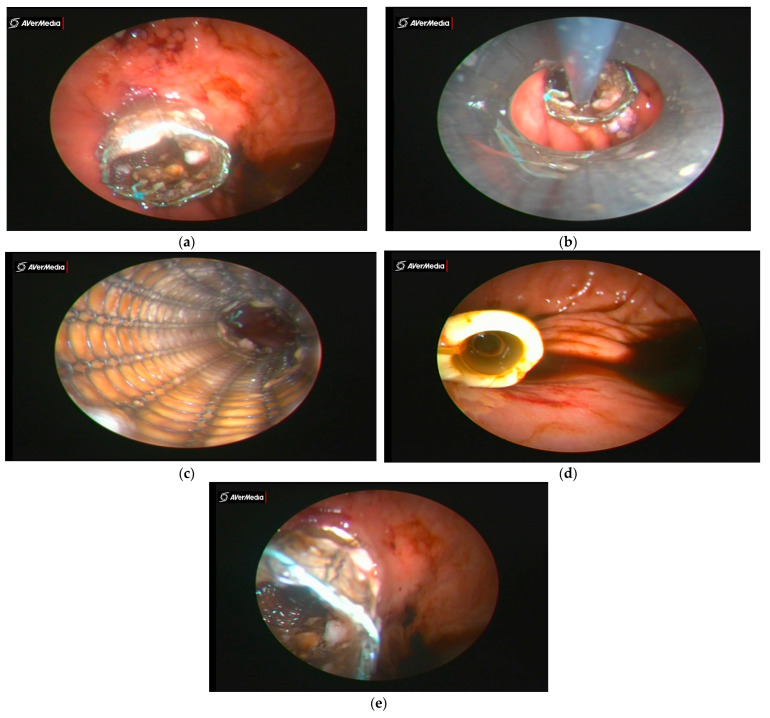
(**a**) Endogastric image of distal end of SEMS prosthesis crossing the cardial orifice. (**b**) Collection of bioptic material after drilling. (**c**) Proximal endoprosthetic view and SEMS. (**d**) Silicone prosthesis distal end—endogastric image. (**e**) Endogastric image of distal end of flexometallic prosthesis through the cardial orifice—tumor at ⅓ of distal esophagus.

**Figure 5 healthcare-11-00815-f005:**
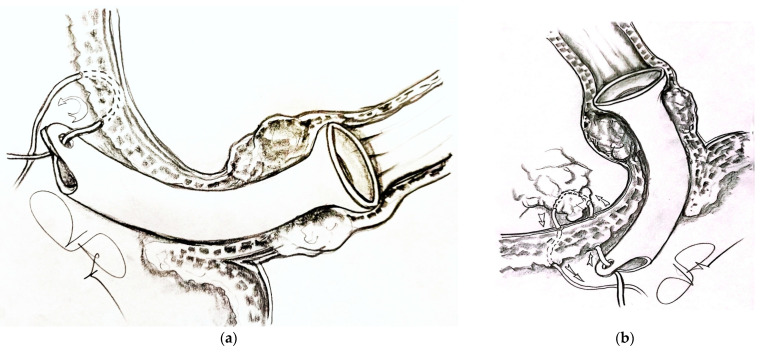
(**a**) Endogastric fixation of stent and (**b**) transparietogastric fixation of stent.

**Table 1 healthcare-11-00815-t001:** Characteristics of study group.

Characteristic *n* = 63	All	Pezzer 35 (48.6)	Silicone 23 (31.9)	SEMS 5 (6.9)	*p*-Value
Gender, *n* (%)					
Male	57 (90.5)	32 (91.4)	20 (87.0)	5 (100.0)	0.805
Female	6 (9.5)	3 (8.6)	3 (13.0)	-	
Age (years, mean ± SD) (range)	65.22 ± 10.46 (36–95)	66.37 ± 9.84 (51–95)	63.78 ± 12.23 (36–92)	63.80 ± 4.91 (58–68)	0.629
Anemia, *n* (%)	46 (73.0)	25 (71.4)	17 (73.9)	4 (80.0)	1.000
Smoking, *n* (%)	60 (95.2)	33 (94.3)	22 (95.7)	5 (100.0)	1.000
Alcohol, *n* (%)	45 (71.4)	22 (62.9)	19 (82.6)	4 (80.0)	0.273
Esophageal cancer					
Squamous	36 (57.1)	10 (28.6)	23 (100)	3 (60.0)	0.000
Adenocarcinoma	27 (42.9)	25 (71.4)	-	2 (40.0)	
Location of cancer, *n* (%)					
Upper esophagus	13 (20.6)	5 (14.3)	7 (30.4)	1 (20.0)	0.000
Middle esophagus	23 (36.5)	5 (14.3)	16 (69.6)	2 (40.0)	
Lower esophagus	27 (42.9)	25 (71.4)	-	2 (40.0)	
Fistula, *n* (%)	8 (12.7)	2 (5.7)	5 (21.7)	1 (20.0)	0.116
Ogilvie dysphagia score (prior stent placement), *n* (%)					
3	25 (39.7)	14 (40.0)	8 (34.8)	3 (60.0)	0.630
4	38 (60.3)	21 (60.0)	15 (65.2)	2 (40.0)	
Ogilvie dysphagia score (after stent placement), *n* (%)					
0	51 (81.0)	30 (85.7)	17 (73.9)	4 (80.0)	0.545
1	12 (19.0)	5 (14.3)	6 (26.1)	1 (20.0)	
Nutritional status, *n* (%)					
Body weight loss	48 (76.2)	29 (82.9)	16 (69.6)	3 (60.0)	0.344
Cachexia	15 (23.8)	6 (17.1)	7 (30.4)	2 (40.0)	
Chemotherapy, *n* (%)	9 (14.3)	3 (8.6)	6 (26.1)	-	0.152
Radiotherapy, *n* (%)	9 (14.3)	2 (5.7)	6 (26.1)	1 (20.0)	0.060
Gastrostomy, *n* (%)					
Pre-operative	6 (9.5)	3 (8.6)	3 (13.0)	-	0.710
Intra-operative	7 (11.1)	3 (8.6)	4 (17.4)	-	
Jejunostomy, *n* (%)	5 (7.9)	2 (5.7)	2 (8.7)	1 (20.0)	0.443
Tracheostomy, *n* (%)	5 (6.9)	2 (5.7)	2 (8.7)	-	1.000

**Table 2 healthcare-11-00815-t002:** Post-stent placement complications.

Complication	All	Pezzer 35 (48.6)	Silicone 23 (31.9)	SEMS 5 (6.9)	*p*-Value
Pain, more than 1 month, n (%)	38 (60.3)	18 (51.4)	16 (69.6)	4 (80.0)	0.274
Hemorrhage, n (%)	3 (4.8)	1 (2.9)	2 (8.7)	-	0.655
Stent migration, n (%)	5 (7.9)	1 (2.9)	2 (8.7)	2 (40.0)	0.038
Restenosis, n (%)	6 (9.5)	3 (8.6)	2 (8.7)	1 (20.0)	0.634

**Table 3 healthcare-11-00815-t003:** Death causes and survival time.

	All	Pezzer 35 (48.6)	Silicone 23 (31.9)	SEMS 5 (6.9)	*p*-Value
Death, n (%)	12	5 (41.7)	6 (50.0)	1 (8.3)	0.545
Death causes					
SEPSIS	4 (33.3)	2 (40.0)	2 (33.3)	-	1.000
Restenosis	2 (16.7)	1 (20.0)	1 (16.7)	-	1.000
Hemorrhage	1 (8.3)	1 (20.0)	-	-	1.000
Cardiac causes	5 (41.7)	2 (40.0)	3 (50.0)	-	0.593
Pulmonary causes	2 (16.7)	-	2 (33.3)	-	0.283
Others	1 (8.3)	-	-	1 (100)	0.079
Survival period, months	10.75 ± 15.72 (0–50)	5.60 ± 10.85 (0–25)	16.50 ± 19.19 (0–50)	2.00	0.504

## Data Availability

The data presented in this study are available upon request from the corresponding author.

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
