# Peer review of "Laparogastroscopy—A Transgastric Laparoscopic Approach for Malignant Esophageal Stenosis"

_healthcare, 2023, doi:10.3390/healthcare11060815_

Round 1
Reviewer 1 Report (Previous Reviewer 2)
Line 232 - "laparogastrscopy" - the typo,
Author Response
Dear Reviewer,
Thank you very much for your kind appreciation of our work.
We have corrected the spelling errors as required.
Reviewer 2 Report (New Reviewer)
In this manuscript, the authors proposed the laparogastrocopy procedure, a mini-invasive, palliative, alternative method of gastrostomy. By laparogastroscopic stenting through transtumoral drilling, endoluminal, it could be expected to solve the problem of oral nutrition in patients with unresectable esophageal cancer, thus avoiding percutaneous feeding. The technique was applied in a group of 63 patients with advanced esophageal carcinoma, admitted between January 2015 and December 2020, in an Emergency County Hospital Sibiu, Romania. Different types of stents were used, which were Pezzer, silicone prosthesis, and self-expanding metal stents. As a result, 8 patients had fistula, which was sealed successfully. In addition, postoperative dysphagia was absent in most patients and minimal in 16.6%, so all patients could initiate oral feeding, improving their nutritional status. The average length of hospitalization for all patients was 9.22±5.05 days. The most frequent local complications were restenosis, stent displacement, and bleeding. The mean survival time was 10.75±15.72 months. In general, the technique presented in the manuscript is interesting and could be useful for use in clinical applications. However, the scientific contribution of the manuscript was not well-discussed and the figures in the manuscript were not well prepared. Therefore, it could not be recommended for publication in the journal in its current form.
Following issues should be addressed:
1. Lines 27-29: “The mean survival time was 10.75±15.72 months. Laparogastroscopic stenting could be a valuable alternative in palliative esophageal cancer surgery, improving the quality of life and nutritional status in patients unsuitable for endoscopic stenting”. The authors should add reference data about mean survival time of some other techniques in palliative esophageal cancer surgery to compare.
2. Figures 1, 2, 3 should be re-organized to make sure that each figure is in only one page. In addition, scale bars should be added in each figure so that readers can easily estimate the size of the used equipment.
3. The scientific contribution of the technique described in the manuscript should be clearly discussed.
4. Figures S1-S4. Brief explanations should be added to each figure caption similar to those added by the authors in Figures S7-S9. The results of these figures must also be discussed in the main text.
Author Response
Dear Reviewer, we have addressed all the issues in the review report very carefully. The article has undergone English service provided by MDPI and we attach the proof of verified English.
Following issues should be addressed:
- Lines 27-29: “The mean survival time was 10.75±15.72 months. Laparogastroscopic stenting could be a valuable alternative in palliative esophageal cancer surgery, improving the quality of life and nutritional status in patients unsuitable for endoscopic stenting”. The authors should add reference data about mean survival time of some other techniques in palliative esophageal cancer surgery to compare.
We added reference data about mean survival time of some other techniques (surgical and endoscopic) lines [329-336]
- Figures 1, 2, 3 should be re-organized to make sure that each figure is in only one page. In addition, scale bars should be added in each figure so that readers can easily estimate the size of the used equipment.
We re-organized the figures. The dimensions of used laparoscopic instruments are described in detail in lines 123-126
- The scientific contribution of the technique described in the manuscript should be clearly discussed.
We have enriched the discussion section lines 292-294, 310-336 and the conclusion section lines 373-377

Reviewer 3 Report (New Reviewer)
With great joy, I read the manuscript: “Laparogastroscopy—a Transgastric Laparoscopic Approach for Malignant Esophageal Stenosis” by Lupu-Petria et al. It is a well-written manuscript in understandable language. The authors report the outcomes of esophageal stent placing in 63 patients with esophageal tumours that cannot be treated with curative intent and that cannot be passed by endoscopy alone. This is a relevant topic and addresses an important issue in the palliative treatment of esophageal cancer. I have a few issues I would like to see resolved:
- What is meant by “social alternative”? line 56 page 2
- How were complications detected? Were there any leakages or intestinal perforations?
- Stent migration in 40% of cases in the SEMS group seems high. Please comment on this.
Author Response
Dear Reviewer,
Thank you very much for your kind words and helpful recommendations. We have revised the manuscript accordingly :
- What is meant by “social alternative”? line 56 page 2
By “social alternative” we meant: The patient can eat orally with the family, as a healthy person, not as a gastrostomised person. Social insertion consists in the ability to eat orally, which is aesthetic, functional.
As a result of the procedure , the patient presereved the the ability to taste food, the ability to chew, to be able to eat in public or with the family, and to avoid overly sophisticated methods of food preparation compared to PEG and surgical gastrostomy.
- How were complications detected? Were there any leakages or intestinal perforations?
Postoperative complications were determined clinically (general condition, dysphagia and dyspnea score were assessed), and on the basis of chest radiography, computed tomography of the chest and abdomen if necessary, abdominal ultrasound, endoscopic ultrasound and endoscopy in case of stent migration or restenosis of stent due to impacted food debris or tumor overgrowth that stenosed the prosthesis.
We had no leakages, intestinal perforation or postoperative fistula by using laparogastroscopic method.
- Stent migration in 40% of cases in the SEMS group seems high. Please comment on this.
[lines 130-139] The stent type was selected on an individual case basis. For the hard cartilaginous tumors, we used rigid plastic stents (Pezzer, silicone), avoiding the transformation of self-expandable metal stents into clepsydra (shape) stents. The rigid plastic stents have good strength, relieve dysphagia almost immediately, and prevent tumor ingrowth. In the case of long tumors in the axis of the esophagus, Pezzer stents are preferred because their length can be adapted intraoperatively. We also used SEMS, which are easier to fix than plastic or semi-rigid stents but are less stable. The uncovered and partially covered SEMS are at high risk of restenosis and the covered ones are prone to migration. Another criteria for not using SEMS was their availability and higher financial costs in comparison with plastic or semirigid stents.
So, we used SEMS in our study in a few cases because they are less stable although they are easier to fix than plastic or semi-rigid stents. We used more covered SEMS which are prone to migration.
Almost all cases were stented with rigid plastic stents because they have good strength, they relieve dysphagia almost immediately and prevent tumor ingrowth. (partial covered SEMS are at high risk of restenosis)
Reviewer 4 Report (New Reviewer)
The level of English in the text requires very careful correction.
In my opinion, this technique has no medical, technical, ethical, or economic advantages.
This method combines all the risks of both endoscopic and laparoscopic approaches. From my point of view, a simple and quick endoscopic or laparoscopic gastrostomy or jejunostomy is much more elegant, and easier, and exposes the patient to minimal risk.
Author Response
R
We regret your point of view.
From our point of view our mini-invasive, palliative method ,laparogastroscopy, is a safe, effective, efficient procedure, a good alternative of PEG or surgical gastrostomy recommended in patients with stage IV esophageal cancer in which endoscopic stenting is impossible because of tumor overgrowth.
Our method is comparable with endoscopic stenting regarding complications, survival, postoperative mortality (we had no perforations, no leakages, no postoperative fistula, preoperative fistula were sealed successfully).
Dysphagia was absent in most patients, minimal in 16,6% percent so all patients could initiate oral feeding improving their nutritional status.
Apparently, in our study, the average length of hospitalization for all patients was high but it is justified by the fact that these patients with stage IV esophageal cancer have an affected general condition, present multiple comorbidities and require additional surveillance.Another reason is that the method is innovative and requires special postoperative control.
Compared to PEG, by using our procedure ,patients that could not be stented by gastroenterologists preserved the ability to taste food, the ability to chew, to eat in public or with their family and avoid overly sophisticated methods of food preparation.
Also, the manuscript is a resubmitted manuscript. It has been professionally checked for English language by MDPI English Editing services. Here we attach a proof.

Round 2
Reviewer 2 Report (New Reviewer)
The authors have addressed all issues pointed out by the Reviewer. The manuscript is now recommended for publication in the journal.
Reviewer 4 Report (New Reviewer)
After this change, the manuscript is understandable. However, from my point of view, I prefer a fast laparoscopic or laparotomic gastro- or jejunostomy. The method described in this paper combines the risk of complications of both laparoscopy and endoscopy or stenting.
This manuscript is a resubmission of an earlier submission. The following is a list of the peer review reports and author responses from that submission.
Round 1
Reviewer 1 Report
The title of the article is not suitable. Please give a good title, which can describe the aim of your study.
Abstract:
“The paper presents an original alternative surgical technique” What do you mean by “original”?
Introduction:
The introduction is too long. Please write only two or three chapters and focus on the new procedure.
What are esotracheal and esobronchial? These are very strange abbreviations.
What is the goal of your study?
Do you want to compare the results of the different endo-stenting procedures?
Methods:
Laparogastroscopy technique description:
Please describe the new procedure without your opinions and the benefits of this procedure, which you can discuss in the discussion. For example:
Line 103-108: These sentences do not belong to the methods and can be written in the discussion.
Line 140: You must write what you believe in the discussion.
Line 140-154: Please write the inclusion criteria and summarize this chapter.
Line 180-183: Are these methods or discussions?
2.2. Patients’ selection in the study group:
How did you define: technical success rate, randez-vous technique
Please mention the types of catheters and stents (Pezzer, Silicon, Flexometal) in the methods and the difference between them. Which criteria did you use in choosing between them?
You must include the full company name of all used catheters and stents.
Results:
Line 235: “9 pactients were excluded from this analysis due to the following causes” How many patients did achieve the inclusion criteria and were included in this study?
Line 239-240: “most cases of esophageal cancer occurring in patients aged 56 - 75 years,” Please write only your results.
Line 247: Please define “primary fistula” in the methods.
There is no statistical comparison between the different stents.
Table 1:
Age: Please use only the range without SD.
What was the technical success rate?
Discussion:
Please summarize your findings in the first chapter of the discussion.
Please explain direct or mediated groove-vacuz technique in the methods.
Lines 344- 361: These chapters include results from your study, which have to be written in the results section. Please distinguish between writing results and discussion.
Where are your conclusions?
There is a lot of chaos in this article. There is no clear presentation of the methods, results, and discussion. Each division contains information from the other.
There are no clear goals in the article.
The statistical analysis is poor and contains only numbers and percentages.
There is no comparison between the different surgical procedures used in the study.
Author Response
Dear Editors, Distinguished Reviewer,
First of all, we want to express our great consideration for the opportunity to revise our manuscript. We highly appreciate the distinguished reviewer’s professionalism, considerations and constructive suggestions to improve of our work. We have tried our best to improve the manuscript, further addressing the questions raised to this goal.
Comments and Suggestions for Authors
The title of the article is not suitable. Please give a good title, which can describe the aim of your study.
A:
We changed the title: Laparogastroscopy - transgastric laparoscopic approach of malignant esophageal stenosis
Abstract:
“The paper presents an original alternative surgical technique” What do you mean by “original”?
A:
The paper presents a mini-invasive, palliative, alternative method of gastrostomy recommended by the gastroenterologist. Laparogastroscopic stenting through endoluminal transtumoral drilling solves the problem of oral nutrition in patients with unresectable esophageal cancer, avoiding percutaneous feeding.
The originality of the method is given by the retrograde approach of the esophageal tumor. The working room is the stomach and the instrument is the laparoscope inserted through the temporary gastrostomy. We did not find a similar procedure in the literature.
Introduction:
The introduction is too long. Please write only two or three chapters and focus on the new procedure.
What are esotracheal and esobronchial? These are very strange abbreviations.
What is the goal of your study?
Do you want to compare the results of the different endo-stenting procedures?
A:
We trimmed the introduction section.
We changed terms esotracheal with “fistula to the trachea” and esobronchial with “fistula to the bronchus”.
The aim of this paper is to describe a mini-invasive, palliative method: laparogastroscopic stenting through transtumoral drilling, endoluminal. This method represents a technical, biological and social alternative to gastrostomy, jejunostomy or pharyngostomy, procedures that are also used in open or laparoscopic surgery for the oncologic patients. Laparogastroscopy is a solution of stenting unresectable esophageal cancer in: (1) cases that cannot be resolved endoscopically due to the technical impossibility of crossing the stenosis (the stenosis is complete and does not allow orthograde passage of the guide wire), (2) tumor stenosis located less than 2 cm from the upper esophageal sphincter, (3) avoidance of the esogastric junction
We did not intend to compare different endo-stenting procedures, the article aim was to describe the laparogastroscopic stenting procedure and different stents used.
Methods:
Laparogastroscopy technique description:
Please describe the new procedure without your opinions and the benefits of this procedure, which you can discuss in the discussion. For example:
Line 103-108: These sentences do not belong to the methods and can be written in the discussion.
Line 140: You must write what you believe in the discussion.
Line 140-154: Please write the inclusion criteria and summarize this chapter.
Line 180-183: Are these methods or discussions?
2.2. Patients’ selection in the study group:
How did you define: technical success rate, randez-vous technique
Please mention the types of catheters and stents (Pezzer, Silicon, Flexometal) in the methods and the difference between them. Which criteria did you use in choosing between them?
You must include the full company name of all used catheters and stents.
A:
We reviewed the sections introduction, methods, results and discussions and conclusion. We hope that the new version of the article makes a proper distinction between these sections.
The cases presented in lines 140-154 were included in supplementary material, referring to them in the discussion and conclusion section. They illustrate the versatility of the method (lines 318-320)
We defined randez-vous technique (lines 82-93), we mentioned the types of catheters and stents in the methods and the reasons why we choose one type or other (lines 102-108), we included the company name of used catheters and stents (lines 99-101).
Results:
Line 235: “9 pactients were excluded from this analysis due to the following causes” How many patients did achieve the inclusion criteria and were included in this study?
Line 239-240: “most cases of esophageal cancer occurring in patients aged 56 - 75 years,” Please write only your results.
Line 247: Please define “primary fistula” in the methods.
There is no statistical comparison between the different stents.
Table 1:
Age: Please use only the range without SD.
What was the technical success rate?
A:
Between 2015 and 2020, a number of 72 patients with esophageal neoplasm in stage III/IV presented at the Emergency Hospital. Of this 72 patients, 9 patients were excluded from this analysis due to different causes (1 patient was out of surgical resources, 2 refused prothesis, 2 had gastrostomy, 1 had gastrointestinal bleeding and 3 required further investigation and treatment) and the rest of 63 patients underwent esophageal stenting for malignant esophageal obstruction through laparogastroscopy and randez-vous technique, and were included in analysis.
By “primary fistula” we wanted to refer to cases of patients presenting with fistula at admission in the clinic.
As we stated, the article aim was to describe / present the laparogastroscopic stenting procedure (lines 53-61) and different stents used; we did not intend to compare different endo-stenting procedures.
In our opinion, we consider that the way of presenting the quantitative variables in the form of average ± standard deviation (range) is one that is often used. We can make the requested modification if you consider it absolutely necessary.
Discussion:
Please summarize your findings in the first chapter of the discussion.
Please explain direct or mediated groove-vacuz technique in the methods.
Lines 344- 361: These chapters include results from your study, which have to be written in the results section. Please distinguish between writing results and discussion.
Where are your conclusions?
There is a lot of chaos in this article. There is no clear presentation of the methods, results, and discussion. Each division contains information from the other.
There are no clear goals in the article.
The statistical analysis is poor and contains only numbers and percentages.
There is no comparison between the different surgical procedures used in the study.
A:
We summarized our findings in the first chapter of the discussion and conclusions section.
We defined randez-vous technique (direct and indirect, lines 82-93) in the method section. The term “groove-vacuz technique” was a writing error.
We reviewed the sections introduction, methods, results, discussions and conclusion. We hope that the new version of the article makes a proper distinction between these sections.
As we stated, we did not intend to compare different endostenting procedures, the article aim was to describe / present the laparogastroscopic stenting procedure (lines 53-61) and different stents used. We decided to leave the lines 285-292 in the discussion and conclusions section because they represent discussions regarding the characteristics of the techniques.

Reviewer 2 Report
Please see the attachment.

Author Response
Dear Editors, Distinguished Reviewer,
First of all, we want to express our great consideration for the opportunity to revise our manuscript. We highly appreciate the distinguished reviewer’s professionalism, considerations and constructive suggestions to improve of our work. We have tried our best to improve the manuscript, further addressing the questions raised to this goal.
The authors presented a novel approach of stenting esophageal tumors from both directions laparo-endoscopically.
The study is mostly descriptive in nature. It would be more convincing to readers as a better method if some statistical comparisons are made with other published results. Demonstrating a statistically significant different in results would improve the value of this approach.
More details on the length of pre-op stricture and the diameters of the stents inserted would also help establish this approach.
The introduction may benefit some streamlining and trimming. The readability may improve with the help of a native English editor.
A:
The aim of this paper is to describe a mini-invasive, palliative method: laparogastroscopic stenting through transtumoral drilling, endoluminal. This method represents a technical, biological and social alternative to gastrostomy, jejunostomy, or pharyngotomy, procedures that are also used in open or laparoscopic surgery for oncologic patients. Laparogastroscopy is a solution of stenting unresectable esophageal cancer in (1) cases that cannot be resolved endoscopically due to the technical impossibility of crossing the stenosis (the stenosis is complete and does not allow orthograde passage of the guide wire), (2) tumor stenosis located less than 2 cm from the upper esophageal sphincter, (3) avoidance of the eso-gastric junction. The originality of the method is given by the retrograde approach of the esophageal tumor. The working room is the stomach and the instrument is the laparoscope inserted through the temporary gastrostomy. We did not find a similar procedure in the literature.
Tumor size was assessed preoperatively and intraoperatively using objective examination, imaging (CT, radiology, endoscopy, and mixed endoscopic and laparoscopic evaluation, with visualization of the cranial and caudal pole of the tumor). Stent sizes were chosen according to the hardness and length of the tumor. Information regarding stents dimension was added in lines 99-108
We reviewed the sections introduction, methods, results, discussions, and conclusion sections. We hope that the new version is more appropriate / improves readability.
Reviewer 3 Report
The authors presented a novel approach of stenting esophageal tumors from both directions laparo-endoscopically.
The study is mostly descriptive in nature. It would be more convincing to readers as a better method if some statistical comparisons are made with other published results. Demonstrating a statistically significant different in results would improve the value of this approach.
More details on the length of pre-op stricture and the diameters of the stents inserted would also help establish this approach.
The introduction may benefit some streamlining and trimming. The readability may improve with the help of a native English editor.
Author Response
Dear Editors, Distinguished Reviewer,
First of all, we want to express our great consideration for the opportunity to revise our manuscript. We highly appreciate the distinguished reviewer’s professionalism, considerations and constructive suggestions to improve of our work. We have tried our best to improve the manuscript, further addressing the questions raised to this goal.
The authors presented an article that describes an interesting technique that is an alternative and a rescue in challenging patients with critical dysphagia. This article could potentially be published but requires a thorough overhaul in terms of style and form.
Below are some of my sample comments:
1) INTRODUCTION - definitely too long! First, use shorter sentences! Secondly, the introduction should outline the main problem and introduce the reader to the subject. Some sentences must be crossed out, and some could be included in the discussion.
We reviewed the introduction section
2) MATERIALS AND METHODS - your review of cases sentences starting with the words "To illustrate ..." in this type of article are unnecessary. You have an interesting method - describe it, describe the effects, but the entire segment should be deleted because it is unnecessary. Alternatively, it can be sent as Supplementary Material.
The cases presented in lines 140-154 were included in supplementary material, referring to them in the discussion and conclusion section. They illustrate the versatility of the method (lines 318-321)
3) Statistical analysis - please add which test was used to check the normality of the distribution.
We added the required informations in lines 167-168
4) RESULTS - the inclusion and exclusion criteria should be removed from the text by presenting them on the flowchart. Also, note the whole Results section: if something is in the table, generally do not repeat it in the text - unless the result is groundbreaking or exciting! Too many numbers and data in the text - tough to read!
We reviewed the result section
5) Discussion - too long, too much repetition of data from your Results, not enough presentations of interesting papers that support your results! Note to the entire article. There are no citations in some places where quite essential statements are made. You should expand on paragraphs like "These indications and type of complications ...", "In our study, the avarage ..." and the entire limitations section. Please, add relevant citations and data sources. I find the paragraph starting with "Two patients ..." and ending with "was removed by coughvomitting effort" redundant and should be removed entirely.
We reviewed the discussion and conclusion section.
The aim of this paper is to describe / present a mini-invasive, palliative method: laparogastroscopic stenting through transtumoral drilling, endoluminal. The originality of the method is given by the retrograde approach of the esophageal tumor. The working room is the stomach and the instrument is the laparoscope inserted through the temporary gastrostomy. We did not find a similar procedure in the literature.
The paragraph starting with "Two patients ..." and ending with "was removed by cough-vomitting effort" illustrate the failures due to the type of pathology, type of location (pharyngoesophageal) and the type of evolution (cranial migration of the tumor). We considered to mention these cases as the limitations of the laparogastroscopic method.
To sum up, the technique and subject matter, from my perspective, as a thoracic surgeon, is very interesting. Congratulations on your effort and great results. By contrast, the article is written in a way that is difficult to read, too much data is repetitive and unnecessary, some segments are way too long - this makes the reader ploughing through the article instead of reading with interest. It requires some thought and refinement.

Round 2
Reviewer 1 Report
The study requires a thorough revision of the English language.
The authors did not respond to many points in the previous review.
The lack of comparison and statistical analysis of different endostents is the main fault of this article.
Lines 65–124: 2.1. Laparogastroscopy technique description: The description of the procedure requires intensive revision and rewriting.
Lines 76–77 “In esogastric placements the stent can be fixed to the small curvature of the stomach, endogastric or anti-migration” Can you explain how you fix the stent to the small curvature.
Line 82 “The randez-vous technique.”: The sentence is incomplete.
Lines 85–87 “ The technique involves a partnership between gastroenterology and surgery and can be used at the extreme level of the digestive tract, i.e. esophagus-stomach and rectum-sigmoid.” This sentence is not suitable to write in the methods. Please remove it.
Which technique did you use in your cases indirect or direct randez-vous technique? If both, how many of both?
Can you rewrite the text in lines 99–108 and clearly describe the criteria for using each type of stents?
Lines 117-124 “Resumption of liquid feeding will be done postoperatively when the post-anesthesia status allows. In the first postoperative day the hydrolactosaccharide diet is recom-mended. Starting from the third postoperative day the semi-solid feeding is permitted, specifying the necessity of its crushing. When switching to a semi-solid diet, the patient will ingest a spoonful of olive oil before the meals to act as a lubricant, and will consume 121 a glass of fizzy drink (Coke) at the end of the meals, for the entire life (for washing the prothesis). Vitamins A, C, E and Resveratrol with antioxidant role, also improve the qual-ity of life of these patients and oncoadjuvant therapy.” What does this text add to the laparogastroscopy technique descriptionis. Please remove it.
Where are the conclusions?
Can you write a take-home message?
Author Response
Dear Editor, Distinguished Reviewer,
We highly appreciate the considerations and constructive suggestions to improve of our work. We have tried our best to improve the manuscript, further addressing the questions raised to this goal.
The study requires a thorough revision of the English language.
The authors did not respond to many points in the previous review.
The lack of comparison and statistical analysis of different endostents is the main fault of this article.
We performed comparative statistical analysis of different endostents and present the statistical significance values in tables.
Lines 65–124: 2.1. Laparogastroscopy technique description: The description of the procedure requires intensive revision and rewriting.
We revisited and rewrote the description of the procedure
Lines 76–77 “In esogastric placements the stent can be fixed to the small curvature of the stomach, endogastric or anti-migration” Can you explain how you fix the stent to the small curvature.
In esogastric placements the stent can be fixed to the small curvature of the stomach, endogastric or transparietogastric (Figure 1d). In endogastric case, a curved needle is used to fix the prothesis to the gastric wall, without penetrating the serosa. The wire is knotted, inside the stomach, using the single port instrument (Figure 3b). In transparietogastric case, a straight needle is used, penetrating the gastric wall from inside to outside. The needle is visualized endoperitoneally and reintroduced into the stomach through a separate penetration, with endogastric knotting of the wire.
We also included two sugestive drawings:
Figure 4. (left) Endogastric fixation of stent (right) transparietogastric fixation of stent
Line 82 “The rendez-vous technique.”: The sentence is incomplete.
We described the technique in lines 97-109.
Lines 85–87 “ The technique involves a partnership between gastroenterology and surgery and can be used at the extreme level of the digestive tract, i.e. esophagus-stomach and rectum-sigmoid.” This sentence is not suitable to write in the methods. Please remove it.
We removed the i.e. sentence.
Which technique did you use in your cases indirect or direct randez-vous technique? If both, how many of both?
Of the 63 patients we used the direct randez-vous technique in 3 cases and the indirect randez-vous technique in 13 cases. 47 patients did not required the randez-vous technique. (lines 180-185).
30 patients were retrogradely laparogastroscopically catheterized. 17 patients were orthogradely catheterized.
Can you rewrite the text in lines 99–108 and clearly describe the criteria for using each type of stents?
Stent type was selected on an individual case basis. For the hard cartilaginous tumors we used rigid plastic stents (Pezzer, silicone) avoiding the transformation of self-expandable metal stents into clepsydra (shape) stents. The rigid plastic stents have good strength, relive dysphagia almost immediately and prevent tumor ingrowth. In case of long tumors, in the axis of the esophagus, Pezzer stents are preferred because their length can be adapted intraoperatively. We also used SEMS which are easier to fix than plastic or semi-rigid stents but are less stable. The uncovered and partially covered SEMS have high risk to restenosis and the covered ones are prone to migration. Another criteria for (not)using SEMS was their availability and higher financial costs in comparison with plastic or semirigid stents.
Lines 117-124 “Resumption of liquid feeding will be done postoperatively when the post-anesthesia status allows. In the first postoperative day the hydrolactosaccharide diet is recom-mended. Starting from the third postoperative day the semi-solid feeding is permitted, specifying the necessity of its crushing. When switching to a semi-solid diet, the patient will ingest a spoonful of olive oil before the meals to act as a lubricant, and will consume 121 a glass of fizzy drink (Coke) at the end of the meals, for the entire life (for washing the prothesis). Vitamins A, C, E and Resveratrol with antioxidant role, also improve the qual-ity of life of these patients and oncoadjuvant therapy.” What does this text add to the laparogastroscopy technique descriptionis. Please remove it.
We removed the indicated lines
Where are the conclusions?
Can you write a take-home message?
We included a conclusion section in lines 292-320 and a take-home message in lines 317-320
Reviewer 2 Report
Dear authors, thank you for correcting the article. The changes made the article much easier to read than its previous version. However, I have a few more comments - they are to make the article in its final version better perceived by the reader.
1. For future papers, please consider using the Shapiro-Wilk test instead of the Kolmogorov-Smirnov test, especially for groups of less than 100 people. The Kolmogorov-Smirnov test can be useful at most for groups of more than 100 people - which has recently been questioned anyway. I recommend using the Shapiro-Wilk test.
My most extensive comments concerns the Discussion:
2. I propose to delete the fragment: "In terms of gender and age, our results are similar to those in the literature, esophageal neoplasm being more common in males than in females and the average age at which 250 this disease is diagnosed is between the 6th and 7th decade of life for both sexes [17,18].". Your article is about a treatment method; this conclusion does not add anything interesting to the Discussion.
3. Lines 258-276, from "In the review article[19]..." - this paragraph is too long, has too many detailed percentages data, it is boring and hard to read. Please shorten this paragraph, select a few of the most important conclusions and cite relevant papers. You should not quote such exact data from other papers - if the reader wants to know them, he will reach for these papers himself!
4. Lines 285-295 from the words "Of the 63 patients..." - I do not understand putting these sentences in the Discussion. In the Discussion, you should confront your results with the results of other scientists. Here you do what you did in the previous version of the paper: you present your own detailed results, just like in the Results section. Please write the conclusion from these few lines and confront it with citations. Alternatively, please remove this fragment.
5. Lines 300-313: I must admit that the part enumerating the advantages of laparogastroscopy is unclear to me. I understand that point (i) is supported by the literature. What about the others from ii - xi? If this is data from the literature, please cite the articles after each advantage. If these are your own conclusions, you can write only those supported by your analysis. You did not make a proper comparison with open surgery in your paper, you did not do a cost-effectiveness analysis, etc. It should be done like this: quote data from the literature, and if something is not the result of your work, you need to remove it. In addition, I ask to use Arabic numbers.
Author Response
Dear Editor, Distinguished Reviewer,
We highly appreciate the considerations and constructive suggestions to improve of our work. We have tried our best to improve the manuscript, further addressing the questions raised to this goal.
Dear authors, thank you for correcting the article. The changes made the article much easier to read than its previous version. However, I have a few more comments - they are to make the article in its final version better perceived by the reader.
- For future papers, please consider using the Shapiro-Wilk test instead of the Kolmogorov-Smirnov test, especially for groups of less than 100 people. The Kolmogorov-Smirnov test can be useful at most for groups of more than 100 people - which has recently been questioned anyway. I recommend using the Shapiro-Wilk test.
We retested the normality of the data using the Shapiro-Wilk test.
My most extensive comments concerns the Discussion:
- I propose to delete the fragment: "In terms of gender and age, our results are similar to those in the literature, esophageal neoplasm being more common in males than in females and the average age at which 250 this disease is diagnosed is between the 6th and 7th decade of life for both sexes [17,18].". Your article is about a treatment method; this conclusion does not add anything interesting to the Discussion.
We deleted the fragment.
- Lines 258-276, from "In the review article[19]..." - this paragraph is too long, has too many detailed percentages data, it is boring and hard to read. Please shorten this paragraph, select a few of the most important conclusions and cite relevant papers. You should not quote such exact data from other papers - if the reader wants to know them, he will reach for these papers himself!
We revised the paragraph.
- Lines 285-295 from the words "Of the 63 patients..." - I do not understand putting these sentences in the Discussion. In the Discussion, you should confront your results with the results of other scientists. Here you do what you did in the previous version of the paper: you present your own detailed results, just like in the Results section. Please write the conclusion from these few lines and confront it with citations. Alternatively, please remove this fragment.
We deleted the fragment.
- Lines 300-313: I must admit that the part enumerating the advantages of laparogastroscopy is unclear to me. I understand that point (i) is supported by the literature. What about the others from ii - xi? If this is data from the literature, please cite the articles after each advantage. If these are your own conclusions, you can write only those supported by your analysis. You did not make a proper comparison with open surgery in your paper, you did not do a cost-effectiveness analysis, etc. It should be done like this: quote data from the literature, and if something is not the result of your work, you need to remove it. In addition, I ask to use Arabic numbers.
We revised the paragraph, in lines 293-309

Reviewer 3 Report
This is a descriptive report of a technique variation. A comparison of the results to the historical or published data would make the approach more convincing to the readers.
Author Response
Dear Editor, Distinguished Reviewer,
We highly appreciate the considerations and constructive suggestions to improve of our work. We have tried our best to improve the manuscript, further addressing the questions raised to this goal.
This is a descriptive report of a technique variation. A comparison of the results to the historical or published data would make the approach more convincing to the readers.
The patient who arrives at stenting by laparogastroscopy presents an advanced stage with direct and indirect comorbidities that have blocked endoscopic stenting. Our sample of patients is included in digestive stoma category (impossible to be stented, from gastroenterological point of view) but, by using laparogastroscopy method, this patients are readmitted to the stented patient category.
The paper presents a mini-invasive, palliative, alternative method of gastrostomy recommended by the gastroenterologist. Laparogastroscopic stenting through endoluminal transtumoral drilling solves the problem of oral nutrition in patients with unresectable esophageal cancer, avoiding percutaneous feeding.
The originality of the method is given by the retrograde approach of the esophageal tumor. The working room is the stomach and the instrument is the laparoscope inserted through the temporary gastrostomy. We did not find a similar procedure in the literature.
